# PFP-WGAN: Protein function prediction by discovering Gene Ontology term correlations with generative adversarial networks

**Seyyede Fatemeh Seyyedsalehi[1,2], Mahdieh Soleymani[1]\*, Hamid R. Rabiee[1]\*, Mohammad R. K. Mofrad[2]**

**1** Department of Computer Engineering, Sharif University of Technology, Tehran, Iran, **2** Department of Mechanical Engineering, University of California Berkeley, Berkeley, California, United States of America

\* Rabiee@sharif.edu (HRR), Soleymani@sharif.edu (MS)

## Abstract

Understanding the functionality of proteins has emerged as a critical problem in recent years due to significant roles of these macro-molecules in biological mechanisms. However, in-laboratory techniques for protein function prediction are not as efficient as methods developed and processed for protein sequencing. While more than 70 million protein sequences are available today, only the functionality of around one percent of them are known. These facts have encouraged researchers to develop computational methods to infer protein functionalities from their sequences. Gene Ontology is the most well-known database for protein functions which has a hierarchical structure, where deeper terms are more determinative and specific. However, the lack of experimentally approved annotations for these specific terms limits the performance of computational methods applied on them. In this work, we propose a method to improve protein function prediction using their sequences by deeply extracting relationships between Gene Ontology terms. To this end, we construct a conditional generative adversarial network which helps to effectively discover and incorporate term correlations in the annotation process. In addition to the baseline algorithms, we compare our method with two recently proposed deep techniques that attempt to utilize Gene Ontology term correlations. Our results confirm the superiority of the proposed method compared to the previous works. Moreover, we demonstrate how our model can effectively help to assign more specific terms to sequences.

## Introduction

Proteins are one of the most important macro-molecule families in biology. Each protein is responsible for one or more functions in biological pathways and discovering these functions leads to a deeper understanding of biological mechanisms. This is also critical in designing different disease treatments. However, available in-laboratory techniques to discover protein functions are expensive and time-consuming. Thus, researchers tend to employ computational techniques which are able to infer protein functionalities from other biological data sources

**Data Availability Statement:** All relevant data are publicly accessible via the following URL: http://git.dml.ir/seyyedsalehi/PFP-WGAN.

**Funding:** This work was supported by Iran National Science Foundation (INSF) [Grant No. 96006077]. The funders had no role in study design, data collection and analysis, decision to publish, or preparation of the manuscript.

**Competing interests:** The authors have declared that no competing interests exist.

including protein structures [1], protein-protein interaction networks [2–4], protein sequences [5], or any combination of them [6–9]. Nowadays, due to improvements in sequencing technologies, a large number of protein sequences are available. The UniProtKB database [10] stores more than 70 million sequences where the functionality of around one percent of them is experimentally approved [5]. Experimental methods for gathering other data sources like interaction networks are more costly and noisier than current sequencing technologies and for the majority of proteins the only available data is their sequence [5]. These facts imply the importance of developing sequence-based computational methods for protein function prediction, which is addressed in the current research.

The previous works on this topic can be divided into three categories [11]: The first category contains alignment-based algorithms which assume that homologous protein sequences have the same functionalities [12–14]. The second group is based on finding specific motifs in sequences. These motifs are functional sites, which are considered as signatures of special functions [11]. The last group includes those methods that are based on machine learning and also are able to extract meaningful and high-level features from raw sequences. The well-known benchmarks for computational methods in protein function prediction, like Critical Assessment of Functional Annotation (CAFA) [15], confirm the superiority of machine learning methods compared to the other categories because machine learning techniques are capable to extract higher level features from raw protein sequences [11].

The most well-known database for annotating proteins is Gene Ontology (GO) [16] which was introduced in 1998 to describe the functionality of genes and their products including proteins. This database includes more than 40000 terms in a Directed Acyclic Graph (DAG). A protein can be assigned to more than one GO term. For example, in SwissProt [17], as the most important annotated subset of UniProtKB, around 71 GO terms are assigned to each human protein on average [5]. Moreover, in the GO structure, every term is a more specific version of its parents and whenever a term is assigned to a protein, all of its parents should also be assigned to it. In this context, protein function prediction can be described as a multi-label classification problem in which the DAG structure of GO imposes a redundancy in the label space. Moreover, there are semantic relations between GO terms which can help to increase the accuracy of an annotating model that incorporates these relations.

Until now, several works that consider the GO term relations in their protein function prediction methods have been introduced [18]. The work in [19] obtains the principal directions in the GO term space by Singular Value Decomposition (SVD) to filter out noisy annotations. CSSAG [20] proposes a greedy hierarchical multi-label classification algorithm which can be used in both tree and DAG structured output spaces. To find the optimal solution, CSSAG searches for the best subgraph in the GO hierarchy. Inspired by the topic modeling studies in the text analysis field, [21] model GO terms as words that are from special topics. In fact, they consider these topics as new representations of functions. In [13], a label space dimensional reduction (LSDR) method which considers both the GO structure and the label distribution is introduced. By incorporating the label distribution in calculating latent representations for GO terms, it is able to consider semantic similarities which can not be necessarily derived from the GO DAG. GO2Vec [22] exploits a graph embedding algorithm, and node2vec [23], tries to obtain a vector representation for each GO term based on the structural information of GO graph. The authors in [22] also apply their method to GOA graph which includes both term-term, from the GO graph, and the term-protein, from the annotation information relations. They use these representations to calculate semantic similarities between GO terms and functional similarities between proteins. Onto2Vec [24] constructs a corpus of axioms based on the GO graph. These axioms describe the hierarchical relations in the GO DAG. It then uses the Word2Vec [25] algorithm to find feature vectors for GO terms by this corpus of

sentences. Onto2Vec finds feature vectors for proteins by adding new axioms describing the annotating relations to the corpus or by a linear combination of feature vectors of terms in the protein's GO annotations.

On the other hand, some researches have focused on introducing new methods for extracting features from raw protein sequences. The work in [26] extends the classical linear discriminant analysis to multi-label problems. They find the best subspace that discriminates samples from different classes and exploit it to obtain feature vectors for protein sequences. The recent success of deep learning algorithms in a large number of applications including bioinformatics [27] motivates researchers to adopt it in the computational protein function prediction. Inspired by the natural language processing concepts [28], use the Word2Vec [25] algorithm to extract a vector representation for protein sequences. They also experimentally show how their method successfully capture meaningful chemical and physical properties of proteins. In [29], the Long-Short-Term-Memory (LSTM) deep network is utilized to extract features from protein sequences and classify them into four functional categories. In addition to the power of deep learning models to extract complex features from input samples, the structure of LSTM allows the model to keep important features across long distances of sequences. However, none of the aforementioned deep models considers label correlations during feature extraction.

DeepGO [30] attempts to incorporate the structural information of the gene ontology to a deep feature extractor by explicitly enforcing the true-path-rule of the GO graph to the output. Their deep network includes an embedding layer followed by convolutional and fully connected layers. At the final step, they define an architecture of maximization layers that considers and propagates the GO structural information in the final results. However, using successive max layers in the final part of the network may not provide sufficient gradient (during the training process) to impose the structural constraints of the GO terms into the network. To utilize GO term correlations during the training process, the work in [31] employs multi-task deep neural networks for protein function prediction. In this architecture, some layers of the network are shared through the tasks, i.e. different GO terms. These layers help to extract more generalized and meaningful features from proteins. However, the loss function of this method is a sum of the prediction loss over all the tasks and does not include any information about the task correlations. In [32], authors claim that the transformer [33] model can extract more relevant features from amino acid sequences compared to convolutional layers. This is because the transformer is able to model all pairwise interactions between amino acids of a protein sequence. They also show by feeding the embedding of GO terms as the input, it is able to extract co-occurrence relations of the true-path-rule and use them for its final prediction.

In this paper, we propose to employ a Generative Adversarial Network (GAN) [34] to improve protein function prediction by simultaneously extracting GO term correlations. GANs were initially introduced for training a deep neural network to produce synthetic samples from a desired distribution [34]. These networks have generally two building blocks, generator and discriminator, which are trained in an adversarial training paradigm. The generator block synthesizes samples of a desired distribution and the discriminator assesses the generator outputs to distinguish them from real samples of the target distribution. During the training process, these two blocks fight against each other until an equilibrium point where the generator can fool the discriminator by its synthetic products. The considerable performance of GANs in the field of image processing [35, 36] motivates researchers to exploit them for other data types including biological ones. Works in [37, 38] use GANs to analyze gene expression profiles and works in [39, 40] attempt to synthesize genes and promoters by GANs. Recently

authors in [41] have been proposed to perform data augmentation to generate synthetic training samples by a GAN to improve a classifier accuracy for annotating proteins.

Here we learn the mapping from the input protein to a binary vector of annotated GO terms by utilizing a conditional generator such that the resulted vector cannot be distinguished from valid annotating vectors by a discriminator. The feedback that is provided by the discriminator during the training process is imposed as a loss function to the deep neural network which is used to predict protein functions. By simultaneous training of the above networks, we learn a customized loss function for the annotating model, by considering available training data.

Moreover, considering term correlations helps to overcome noisy annotations that may corrupt the performance of a prediction model. We show that the proposed method is able to model co-occurrence relations that are not necessarily available in the current DAG model between GO terms. An important issue of computational prediction of protein functions is the shortage of positive samples for terms in the deeper levels of the GO DAG which are more specific and informative. Thus, considering semantic similarities is more critical for deeper GO terms. The proposed method achieves higher accuracy compared to the existing methods with the same number of training data and decreases the sample complexity of the problem. We demonstrate that the distance between the proposed and previous methods increases when moving through deeper terms which confirms the importance of incorporating the semantic and architectural similarities for deeper GO terms.

## Materials and methods

In the proposed method, functionalities of a protein are described as a binary assignment vector whose elements show whether or not a protein is responsible for a GO term. Without the loss of generality, we can describe all correlations between GO terms as a joint distribution over the assignment vector of all proteins. For instance, let us consider it is impossible for a protein to be responsible for two special GO terms simultaneously. Then, the probability of assignment vectors in which the elements corresponding to these two special terms are active at the same time is zero. The proposed model learns a joint distribution over the function assignment vectors given the input protein sequence. It helps to extract semantic relations between GO terms for special sequence patterns. Hence, our model is capable of extracting more complicated relations. In the following subsections, we show how we learn the conditional joint distributions and utilize them to annotate protein sequences.

Let $\mathbf{x}$ denote a protein sample that contains either hand-crafted features of a protein sequence or a raw protein sequence itself and $\mathbf{y} \in \mathbb{R}^c$ denote an assignment vector where $c$ shows the number of GO terms. We denote the conditional distribution over assignment vectors $\mathbf{y}$, conditioned on the protein $\mathbf{x}$ distribution over the GO term assignments, by $\mathbf{p}_{GO}(\mathbf{y}|\mathbf{x})$. We define a distribution $\mathbf{p}_m(\mathbf{y}|\mathbf{x})$ that is modeled by a deep neural network to estimate $\mathbf{p}_{GO}(\mathbf{y}|\mathbf{x})$.

### Wasserstein generative adversarial network

Motivated by the success of GAN networks, many extensions have been introduced. Here, we adopt the Wasserstein Generative Adversarial Network (WGAN) [42]. Indeed, considering the arguments of [42] into account, we believe WGAN has better performance for learning a distribution over our discrete space, i.e. the space of all function assignment vectors. The loss function of a WGAN is defined as follows:

$$\arg \min_{G} \arg \max_{D \in L^{(1)}} \mathbb{E}_{\mathbf{y} \sim \mathbf{p}_r}[D(\mathbf{y})] - \mathbb{E}_{\mathbf{y} \sim \mathbf{p}_m}[D(\mathbf{y})], \tag{1}$$

where $\mathbf{p}_r$ is the desired distribution to be modeled, $\mathbf{p}_m$ is the distribution trained by the generator $G$, and $\mathrm{L}^{(1)}$ shows the family of 1-Lipschitz functions. Moreover, $D$ and $G$ denote the discriminator and generator networks, respectively. The term inside the argmin measures implicitly the distance between the two distributions $\mathbf{p}_r$ and $\mathbf{p}_m$ by utilizing the discriminator network. Hence, the model attempts to find a generator with distribution $\mathbf{p}_m$ that provides a good estimate for $\mathbf{p}_r$.

Since we are trying to find the function assignment vector given the input protein sequence, inspired by the idea of conditional GAN [43], we design a conditional generator and a conditional discriminator. The real distribution that we try to learn is the distribution over assignment vectors conditioned on the protein sample, which is denoted by $\mathbf{p}_{GO}(\mathbf{y}|\mathbf{x})$. The discriminator also takes the protein sequence and a function assignment vector (that can be a generated vector by the generator network or the real target vector for the protein sequence) and distinguishes whether this vector is real or fake (i.e. a generated one). Therefore, the loss function of the conditional WGAN for this problem can be defined as follows:

$$
\arg \min_{G} \arg \max_{D \in \mathrm{L}^{(1)}} \mathbb{E}_{\mathbf{x} \sim \mathbf{p}_{\mathbf{x}}} \big[\, \mathbb{E}_{\mathbf{y} \sim \mathbf{p}_{GO}(\mathbf{y}|\mathbf{x})}[D(\mathbf{x}, \mathbf{y})] -
$$
$$
\mathbb{E}_{\mathbf{y} \sim \mathbf{p}_m(\mathbf{y}|\mathbf{x})}[D(\mathbf{x}, \mathbf{y})] \,\big]
$$

(2)

Eq (2) shows the general conditional WGAN loss function. In the following subsections, we explain structures and detailed loss functions of the generator for the protein function prediction problem. The proposed method is called **PFP-WGAN**, since the Protein Function Prediction is accomplished by a conditional WGAN in our method.

## Generator structure and loss function

The generator structure which we use for assigning functions to raw protein sequences directly, is depicted in Fig 1. The raw sequence is represented as a set of 8000 dimensional one-hot vectors to the model. These vectors are too sparse. Therefore, we put an embedding function at the first layer of the generator which converts the one-hot input into a vector of length 128. To have a stochastic generator we add a dropout layer with the rate of 0.2. It also decreases the probability of over-fitting.

For the next layer, we use 32 one-dimensional convolution filters which extract meaningful patterns from the sequence of amino acids. After the training process, each filter is responsible for detecting a specific pattern. By patterns we mean the existence of special sequence of amino acids in specific positions. Meaningful patterns are those which are correlated with different GO terms. When a sequence is passed through these filters an activation map, which shows the matching score between patterns and input sequence, is obtained. These filters are followed by a LeakyReLU activation function. To keep the resulted activation maps smaller and more manageable, they are then passed from an average-pooling layer with the filter size of 64 and stride of 32. We then send activation maps through two fully connected layers. These layers learn nonlinear functions of activation maps. If the model has been trained successfully, outputs of these functions are evidence of biochemical and biophysical features of a protein which are related to its functionalities and the generator is able to annotate protein sequences according to them. The size of the last layer is equal to the number of GO terms and this layer shows the resulting assignment vector. A Tanh activation function is utilized in this layer.

We also compare our method with a recent method that does not work on the raw sequence of proteins. There, we use hand-crafted features of sequences obtained by experts (similarly as in the compared method) and extract protein functionalities from them. Therefore, in this scenario, the generator takes hand-crafted features and without using embedding and

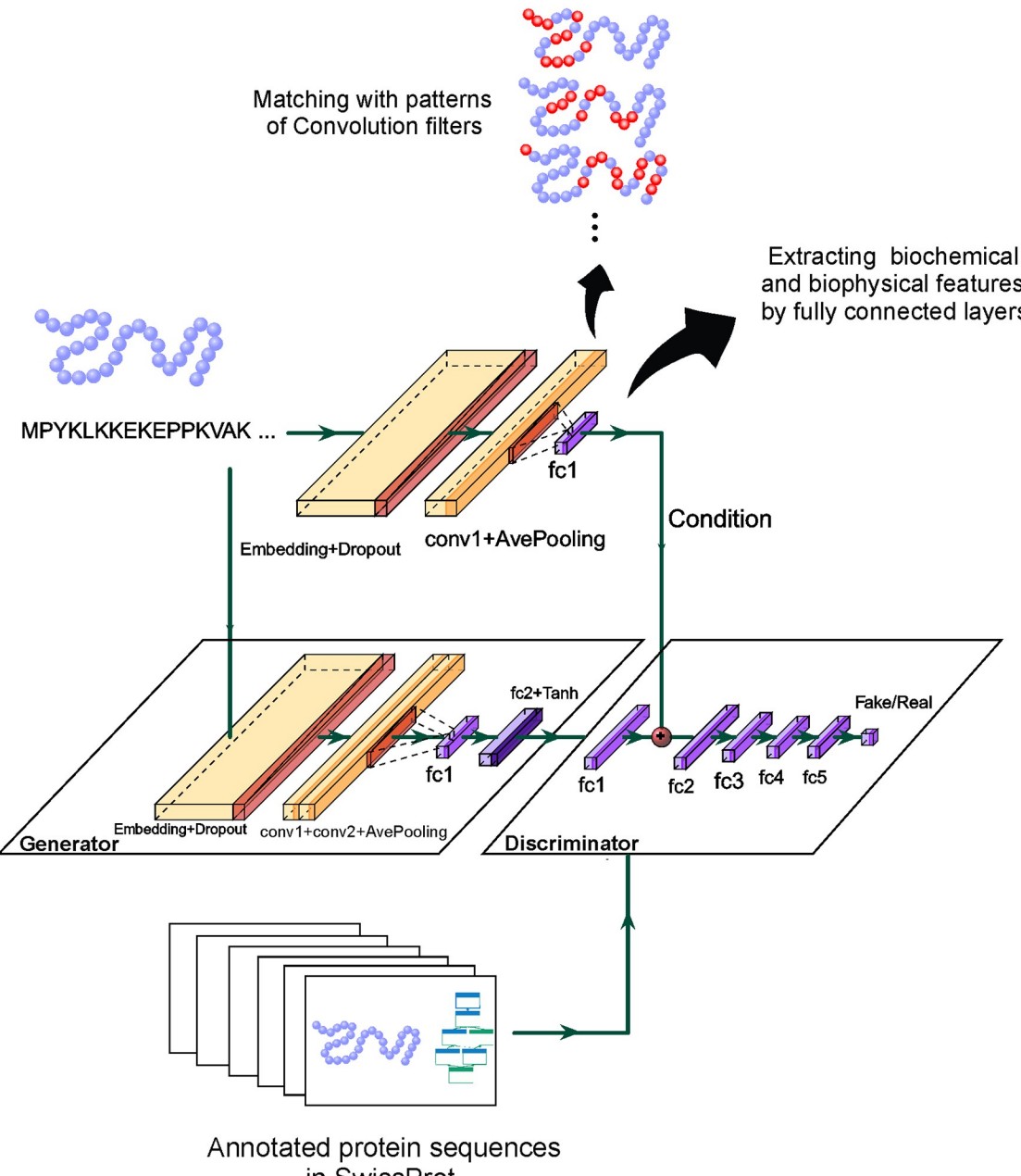

**Fig 1. The proposed method for protein function prediction.** Given the input sequence of amino acids, the generator has an embedding as its first layer which converts one-hot vectors to more compact representations. Then, one dimensional convolution filters are employed to explore meaningful sequential patterns. Then biochemical and biophysical features of the input are extracted from obtained activation maps. These features are used to predict GO annotations by the last fully connected layer. A discriminator judges about the validity of the obtained annotation for this sequence by observing pairs of protein sequences and their experimentally approved annotations in SwissProt. By observing proteins which are annotated by a common set of GO terms, discriminator could extract correlations between GO terms.

convolutional layers yields the function assignment vector. This generator includes 3 fully connected layers with the LeakyReLU activation function and another fully connected layer with the Tanh activation function to find the output.

In order to train the generator, we use the following loss function:

$$\underset{G}{\mathrm{argmin}}\ \mathbb{E}_{\mathbf{x}\sim\mathbf{p_x}}[\mathbb{E}_{\mathbf{y}\sim\mathbf{p}_{GO(\mathbf{y}|\mathbf{x})}}[\mathcal{L}(\mathbf{y}, D(G(\mathbf{x})))] - \lambda_1 D(\mathbf{x}, G(\mathbf{x}))],\tag{3}$$

where the first term is the binary cross entropy loss function which directly compares the generator output for a special sample with its ground truth. The second term is the Wasserstein loss for the generator which is equal to the first term in Eq (2), since $\mathbf{y} \sim \mathbf{p}_m(\mathbf{y}|\mathbf{x})$ shows the generator output and we can replace it with $G(\mathbf{x})$. Finally, $\lambda_1$ is a hyper-parameter which is set to 0.03 in the first experiment and 0.00001 in the second experiment that is chosen according to the performance on the validation set.

## Discriminator structure and loss function

We use two sets of real and fake pairs for training our conditional discriminator. The first one is the real set that includes pairs of input **x** and the corresponding vector assignment in training data. The second set includes fake pairs which consists of **x** and the corresponding generator's output. The discriminator structure is shown in Fig 1. We extract a feature vector from raw sequences **x** by adding embedding, convolutional, max-pooling, and fully connected layers (exactly the same as the first four layers of the generator) to fulfill the condition for the discriminator. In the discriminator network, we first use a fully connected layer to extract features from assignment vectors. The discriminator then concatenates this feature vector with the prepared condition (from the input) and sends them through 5 fully connected layers. The last layer involves a single neuron which scores (protein,function) pairs to distinguish between fake or real ones. The discriminator loss function is formulated as:

$$\underset{D}{\mathrm{argmin}}\ \mathbb{E}_{\mathbf{x}\sim p_{\mathbf{x}}}\left[\ D(\mathbf{x}, G(\mathbf{x})) - \mathbb{E}_{\mathbf{y}\sim\mathbf{p}_{GO(\mathbf{y}|\mathbf{x})}}[D(\mathbf{x}, \mathbf{y})]\ \right]$$
$$+\lambda_2\ \mathbb{E}_{(\tilde{\mathbf{x}},\tilde{\mathbf{y}})\in\tilde{\mathbf{p}}}\left[(\|\nabla_{\tilde{\mathbf{y}}}D(\tilde{\mathbf{x}}, \tilde{\mathbf{y}})\|_2 - 1)^2\right],\tag{4}$$

where the first line is the Wasserstein loss for the discriminator that is obtained from Eq (2). In Eq (4), we omit the constraint of $\mathrm{L}^{(1)}$ from the search space of $D(.)$ and replace it by the term in the second line of Eq (4) that is proposed by [44]. This term is a gradient penalty which keeps the gradient norm of the discriminator around 1. The pairs $(\tilde{\mathbf{x}}, \tilde{\mathbf{y}})$ are produced by weighted averaging (random weights from a uniform distribution) of real and fake pairs and $\tilde{\mathbf{p}}$ describes the resulted distribution. Finally, $\lambda_2$ is a hyper-parameter which is set to 10 in both experiments that is chosen according to the performance on validation set.

## Training and optimization

We train the generator and the discriminator networks alternatively by a training ratio of 10. It means for each iteration of training the generator, the discriminator training is achieved by 10 iterations. The loss functions are optimized by an Adam optimizer with the learning rate of 0.00001, and 20% of the training data is used for validation. Thus, we find the network's weights by 80% of data and then evaluate the resulted model by the remaining 20% to tune the hyper-parameters. The algorithm has been implemented by the Keras deep learning library and trained and tested on a Nvidia gpu GeForce GTX 1080 Ti system.

## Dataset and data representation

We report our results via testing on two different datasets. In the first experiment, we use the data gathered and filtered by [30] in which protein sequences are obtained from SwissProt downloaded on 2016-01. In this dataset, sequences with the length greater than 1002 and ambiguous amino acids are filtered out and sequences with the initial length that is less than 1002 are padded with zeros. In addition, similar to [30], we just keep annotated sequences with experimental evidence code (EXP, IDA, IPI, IMP, IGI, IEP, TAS, IC). The GO terms [16] are downloaded in OBO format http://geneontology.org/page/download-ontology on 2016-01 and similar to [30], terms with less than 250, 50, and 50 annotated proteins for each group of biological processes (BP), molecular functions (MF), and cellular components (CC) are omitted, respectively. This results in 932, 589, and 436 terms in each group. Finally, proteins are randomly divided into training (80%) and test (20%) sets. To represent raw sequences to our model (Fig 1), we divide them into trigrams of amino acids with the overlapping size of two. Considering a dictionary of all possible trigrams, we can show each trigram with a one-hot vector with length 8000. Therefore, each protein sequence is presented by 1000 vectors of length 8000.

The second dataset is identical to the one used in FFPred3 [45]. Protein sequences are from SwissProt's version 2015-5 which are encoded to 258 features including 14 structural and functional aspects. The GO terms are downloaded on 2015-02 and include 605 BP terms, 158 MF terms, and 102 CC terms.

## Evaluation measures

The main measure used to assess different methods is protein-centric $F_{max}$ which was utilized in CAFA challenge [15] and in many recent related works [30, 31]. To calculate it, we define 100 different thresholds $t \in [0, 1]$. Then, for each protein and threshold $t$, we obtain the number of labels truly assigned to the protein ($tp$), the number of protein's labels which are not assigned to it by the model ($fn$), and the number of labels which are falsely assigned to this protein by the model ($fp$). Then, the precision and recall are calculated as follows:

$$\text{Precision}_t = \frac{tp_t}{tp_t + fp_t}, \tag{5}$$

$$\text{Recall}_t = \frac{tp_t}{tp_t + fn_t}, \tag{6}$$

We average the above measures among all proteins to obtain $\text{AvePr}_t$ and $\text{AveRe}_t$ in each interval. Finally, $F_{max}$ is calculated as follows:

$$F_{max} = \max_t \left\{ \frac{2 \times \text{AvePr}_t \times \text{AveRe}_t}{\text{AvePr}_t + \text{AveRe}_t}, \right\}. \tag{7}$$

The other measure which we use is term-centric F1 which is calculated for each label, separately. By defining $tp$ as the number of samples truly assigned to a label, $fn$ as the number of samples which are wrongly not assigned to that label, and $fp$ as the number of samples wrongly assigned to it, we then use Eqs (5) and (6), to calculate the term-centric F1 as:

$$F1 = \max_t \left\{ \frac{2 \times \text{Precision}_t \times \text{Recall}_t}{\text{Precision}_t + \text{Recall}_t} \right\}. \tag{8}$$

We also use three other term-centric measures which are useful for evaluating methods on imbalanced classification problems in which the number of available training samples in one class is much less than the number of samples in another class. The first measure is the Area Under Precision Recall (AUPR) that is obtained for each label as follows:

$$\text{AUPR} = \int_{-\infty}^{\infty} \text{Precision}_t \times \text{Recall}_t' \, dt. \tag{9}$$

Then, we average these values through all the labels and report it. The second and third ones are the Area Under ROC Curve (AUC-ROC) and the Mathews Correlation Coefficient (MCC) which are computed as in [30].

Finally, to check the consistency of results with the true-path-rule, we define the TPR score. We calculate this measure as follows:

$$\text{TPR} = \frac{1}{N} \sum_{n=1}^{N} \sum_{t_i \in \text{annot}(p_n)} \text{card}(\text{anc}(t_i) - [\text{anc}(t_i) \cap \text{annot}(p_n)]) \tag{10}$$

where card(.) shows the cardinality of a set, annot(.) is the set of terms in the annotation of a protein and anc(.) is the set of ancestors of a GO term. According to the true path rule of GO annotations, when a protein is annotated with a GO term, it should also be annotated with the corresponding ancestor terms. A conflict occurs when a protein is not annotated by one of the ancestors of the terms in its annotation. The TPR score calculates the expected number of conflicts in the annotation of a protein.

## Results and discussions

### Experiment 1

In this section, we evaluate PFP-WGAN on the first data set used in [30], and employ similar settings as those used in [30]. The end-to-end structure of the proposed model enables us to extract features from protein sequences and learning GO term correlations, simultaneously. Here, we utilize raw amino-acid sequences as the input to the model. In this part, we compare PFP-WGAN with BLAST [12] and DeepGO-Seq [30]. Setting of these algorithms are exactly as in [30]. DeepGO [30] utilizes a deep network to extract features from protein sequences and protein-protein interaction (PPI) networks and finds the proteins' functions. Authors in [30] attempt to incorporate the structural information about GO by adding a maximization layer which explicitly enforce the "true path rule" at the final step of the network. However, for most of the known protein sequences, the information of protein-protein interaction network is not available. They discuss that in this situation one can use the PPI information of the most similar protein to the query sequence which is found by BLAST [12]. However, they did not report their method's performance for this situation. In addition, this approach limits the algorithm to predict functions of only those sequences for which there is a sufficiently similar protein among those having the PPI information. Here, the DeepGO-Seq as a version of DeepGO that just uses protein sequences to extract functions [30] is compared with our method which only utilizes protein sequences as input. The results of BLAST [12], DeepGO-Seq [30], and PFP-WGAN are compared in Fig 2. In all three branches of GO, PFP-WGAN has better performance compared to BLAST and DeepGO-Seq. Despite the strength of deep network to extract features, in Biological Process (BP) and Molecular Function (MF) branches, DeepGO-Seq performs worse than BLAST. However, as shown in Fig 2, PFP-WGAN obtains a better $F_{max}$ value than the other two competing algorithms. It is worth to mention that the proposed

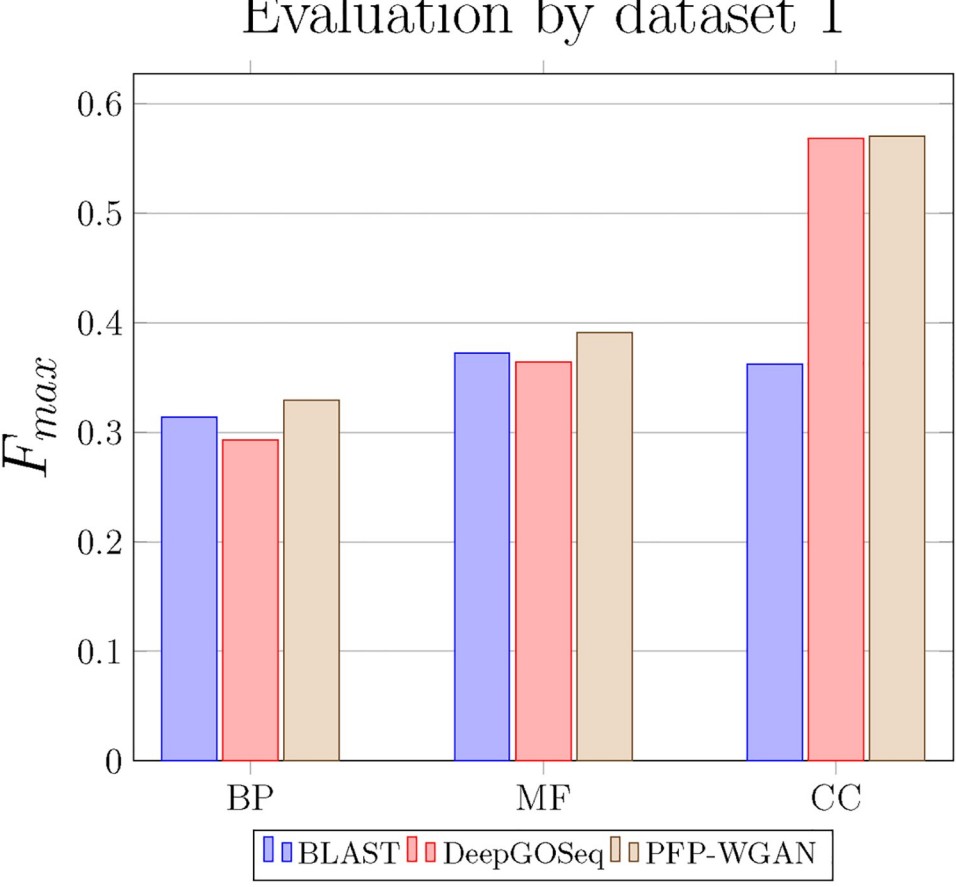

**Fig 2. Comparison of BLAST, DeepGO-Seq and PFP-WGAN on dataset 1.** The $F_{max}$ measure shows the superiority of PFP-WGAN in all three parts of the GO.

method, as opposed to [30], does not employ any additional knowledge (like GO DAG) and automatically discover the relations between different labels (i.e. GO terms).

Table 1 shows the comparison of DeepGO-Seq and PFP-WGAN with three term-centric measures. PFP-WGAN shows better performance in all situations. Considering the fact that both methods utilize a deep network, this result confirms that our discriminator block can effectively extract the GO term correlations and impose them to the generator in order to find more accurate annotations.

Table 2 compares the average prediction time that each of DeepGO-Seq and PFP-WGAN needs. There is not a considerable difference between prediction times. However as the number of terms is increased, the growth of averaged prediction time in DeepGO is more than

**Table 1. Three term-centric measures suitable for unbalanced data.**

| Method | BP | | | MF | | | CC | | |
|---|---|---|---|---|---|---|---|---|---|
| | AUPR | AUC | MCC | AUPR | AUC | MCC | AUPR | AUC | MCC |
| DeepGO-Seq | 0.232 | 0.82 | 0.269 | 0.28 | 0.88 | 0.336 | 0.522 | 0.926 | 0.519 |
| PFP-WGAN | **0.241** | **0.830** | **0.281** | **0.302** | **0.891** | **0.347** | **0.535** | **0.932** | **0.524** |

**Table 2. Average prediction time in seconds for 1000 sequences.**

| Method | BP (932) | MF (589) | CC (436) |
|---|---|---|---|
| DeepGO-Seq | 1.01 | 0.67 | 0.45 |
| PFP-WGAN | 0.96 | 0.84 | 0.73 |

PFP-WGAN. So PFP-WGAN is more scalable for predicting a large number of GO terms simultaneously.

We also calculate F1 for each GO term and average them through functions in each height of the GO graph. The differences between these averages for PFP-WGAN and DeepGO-Seq as a function of the height of terms in the GO graph are presented in Fig 3. As a general trend, we can observe differences between the results of these two methods increase when going through deeper terms. This confirms our intuition about incorporating structural information between output variables (GO terms) during the learning process. Interestingly, terms in higher levels show general functions and positive samples of all their child terms can also be considered as the positive samples of themselves too. In the GO DAG, there is no further valuable correlation between the terms and their child terms which can help to increase the accuracy. Nonetheless, deeper terms can make complicated relations with non-descendant and non-ancestors terms in the GO graph. Thus, extracting correlations between deeper terms is more informative and has higher impact on the classification accuracy. A main bottleneck of deeper terms in the GO graph is the shortage of positive samples which limits the performance of prediction models for these important terms. Fig 4 shows F1 measures for GO terms as a function of positive training samples. The improvement which is obtained by PFP-WGAN for rare terms is more considerable comparing to terms with large numbers of positive samples. It confirms that by incorporating the GO term correlations (more general than the GO DAG) we can compensate this shortage and obtain a better accuracy. Finally, the TPR score of the PFP-WGAN on this dataset is 0.78, 0.3 and 0.11 for BP, MF and CC branches respectively. In addition the total number of grandchild and ancestor pairs of the tree of each branch is 8323, 3266 and 3106.

## Experiment 2

Here, we compare PFP-WGAN with a recently proposed multi-task deep neural network for protein function prediction, MTDNN [31], and a shadow and greedy hierarchical multi-label

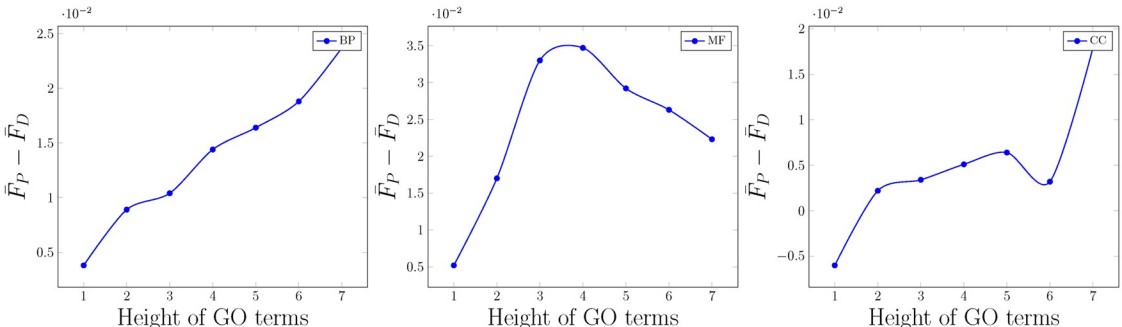

**Fig 3. Differences between average F1 obtained for PFP-WGAN and DeepGO-Seq for the GO terms in each height ($\bar{F}_P$ and $\bar{F}_D$).** In the BP branch (as the most important part of GO with a large number of terms) differences are increased when moving through the deeper terms. In the most and half parts of the charts for CC and MF branches we can observe this pattern too.

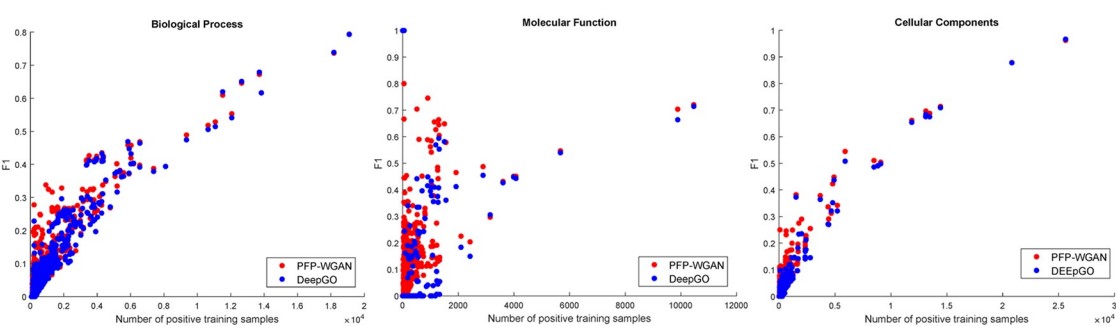

**Fig 4. F1 obtained for PFP-WGAN and DeepGO-Seq for GO terms as a function of number of available positive training samples.** The improvement which is obtained by PFP-WGAN for rare terms is more considerable comparing to terms with large numbers of positive samples.

classification strategy called CSSAG [20]. We evaluate the performance of algorithms on the dataset introduced in FFPred [45]. This dataset includes 258 sequence-derived features of each protein sample and maps them to 605 BP, 158 MF, and 102 CC GO terms. The obtained $F_{max}$ measure for PFP-WGAN, MTDNN, CSSAG and 3 baseline algorithms is shown in Fig 5.

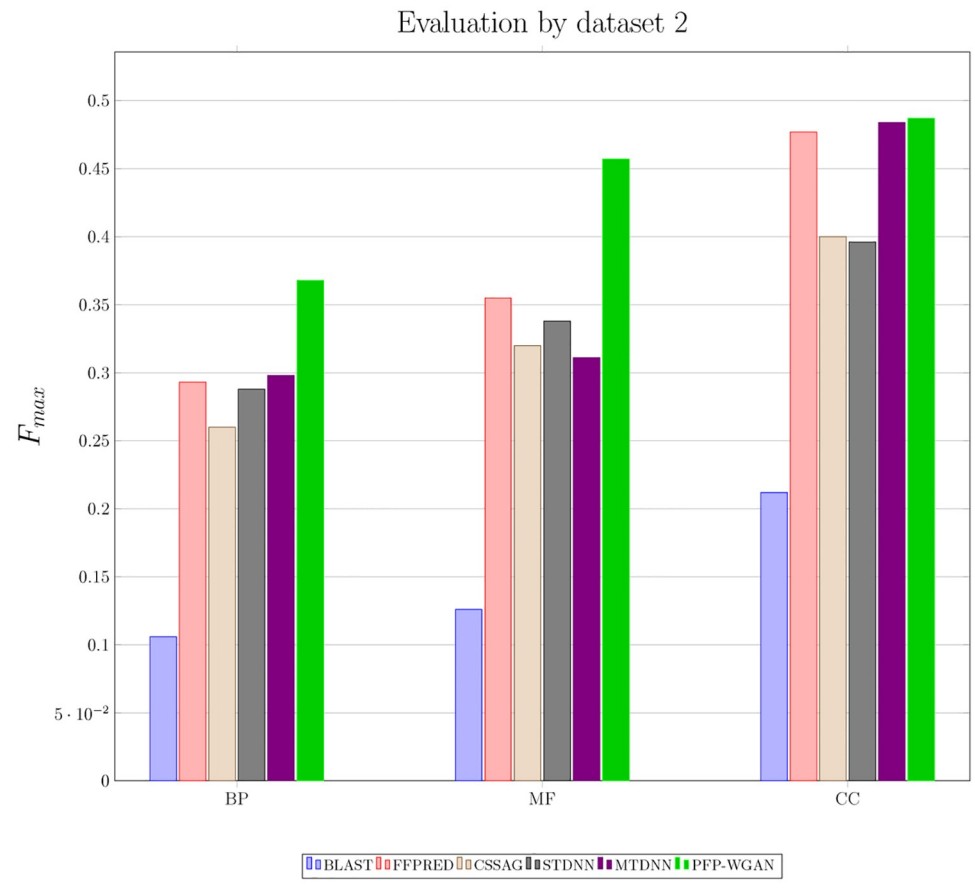

**Fig 5. Comparison of BLAST, FFPRED, CSSAG, STDNN, MTDNN and PFP-WGAN on dataset 2.** The $F_{max}$ measure shows the superiority of PFP-WGAN in all three parts of the GO.

**Table 3. MSE of PFP-WGAN and PFP-S against grandtruth.**

|            | BP       | MF       | CC       |
|------------|----------|----------|----------|
| PFP-S      | 0.46     | 0.41     | 0.51     |
| PFP-WGAN   | **0.38** | **0.39** | **0.46** |

BLAST and FFPred are the first two baseline algorithms. STDNN uses a single fully connected feedforward deep neural network for each GO term separately. Results of baselines and MTDNN are reported from [31]. As shown in Fig 5, the proposed PFP-WGAN has the highest score in all three BP, MF and CC domains.

We also calculate a binary heatmap from results of PFP-WGAN and PFP-S. PFP-S is obtained by omitting the discriminator block, which is responsible for extracting GO term correlations, from PFP-WGAN. For each of these annotation results, this heatmap shows whether each pair of GO terms appear in at least one protein simultaneously or not. We also calculate this heatmap for the training data as the grandtruth which shows two GO terms are as consistent as that a protein can be annotated by both of them simultaneously. Table 3 compares the heatmap of PFP-WGAN and PFP-S against the grandtruth by the mean squared error (MSE). The MSE of PFP-WGAN is less than the MSE of PFP-S in all three branches. This fact confirms that PFP-WGAN has more ability to explore and utilize such relations from the training data.

Fig 6 shows the sensitivity of the PFP-WGAN on parameter $\lambda_1$. This parameter is chosen according to three measures $F_{max}$, micro $F1$ and averaging $F1$ through all terms on validation data. We sum these measures to obtain $F$ and find best value for $\lambda_1$.

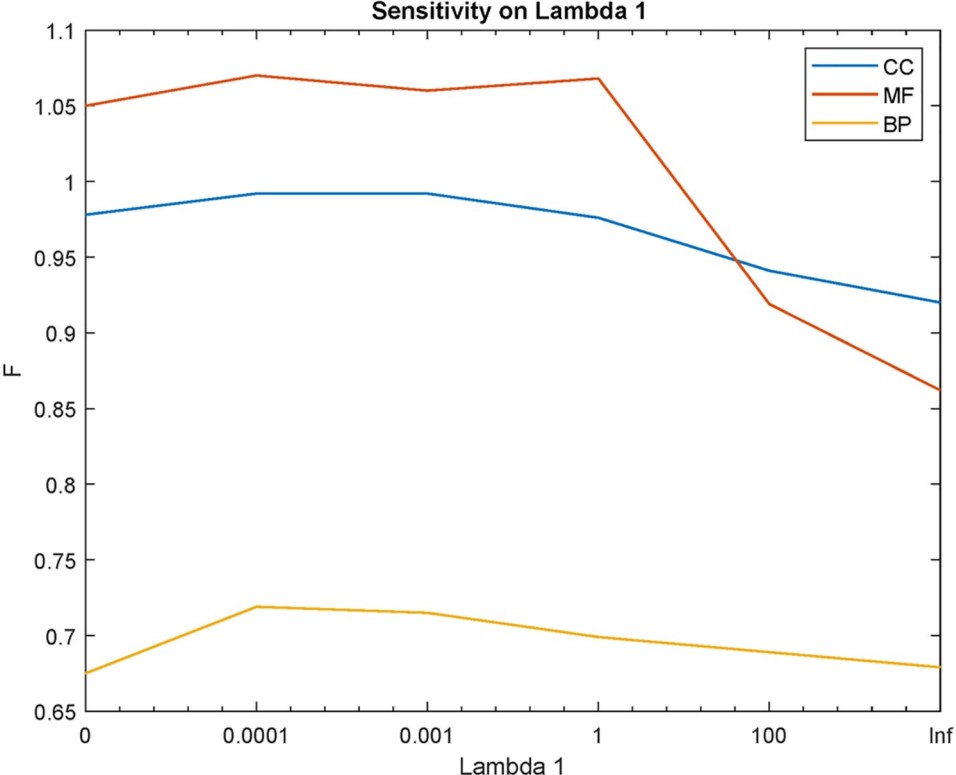

**Fig 6. Sensitivity of the PFP-WGAN on parameter $\lambda_1$.**

Finally, the TPR score of the PFP-WGAN on this dataset is 0.55, 0.11 and 0.13 for BP, MF and CC branches respectively. In addition the total number of grandchild and ancestor pairs of the tree of each branch is 292, 84 and 44.

## Conclusion

As a consequence of improvements in high throughout sequencing technologies, we are faced with a large number of protein sequences in the nature about which there is no other available knowledge. This fact increases the importance of developing techniques to determine these protein's functionalities just with their sequences. An early introduced track of designing such methods is based on finding the most similar protein in a database with known annotations to a query sequence [11] and assigning functions of the retrieved protein to the query. A main limitation of such algorithms occurs for sequences for which we do not have an adequately similar protein in the database. Thus, another trend is based on proposing algorithms which are able to extract biologically meaningful features form a sequence. Deep networks are the most powerful among the currently known models for feature extraction. Here, we propose a new deep architecture for protein function prediction, which uses the protein sequences only and is able to increase the annotation accuracy. The main strength of our model is its ability to extract function correlations and impose them to the annotating process. To the best of our knowledge, this is the first time that a conditional GAN architecture is used to improve the accuracy of a multi-label classification problem. Another advantage of our model corresponds to deeper terms in the GO graph. By extracting term correlations, we are able to decrease the sample complexity of deep terms and obtain higher accuracy. Therefore. we can find more detailed and specific annotations for proteins. The main drawback of our proposed model is that it requires relatively more computational resources, similar to other deep networks.

Our future work include trying to interpret extracted features by our deep network, which would help us to annotate proteins. Considering the ability of deep models in feature extraction we hope to find important biochemical and biophysical meaningful features.

## Acknowledgments

Authors would like to thank Amirali Moeinfar for his help to implement the Wasserstein GAN architecture.

## Author Contributions

**Conceptualization:** Seyyede Fatemeh Seyyedsalehi, Mahdieh Soleymani, Hamid R. Rabiee, Mohammad R. K. Mofrad.

**Formal analysis:** Seyyede Fatemeh Seyyedsalehi.

**Investigation:** Seyyede Fatemeh Seyyedsalehi.

**Methodology:** Seyyede Fatemeh Seyyedsalehi, Hamid R. Rabiee.

**Supervision:** Mahdieh Soleymani, Hamid R. Rabiee, Mohammad R. K. Mofrad.

**Writing – original draft:** Seyyede Fatemeh Seyyedsalehi.

**Writing – review & editing:** Mahdieh Soleymani, Hamid R. Rabiee, Mohammad R. K. Mofrad.

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
