## [Decision Letter · Decision Letter 0]

23 Jun 2020

PONE-D-20-05418

PFP-WGAN: Protein Function Prediction by Discovering Gene Ontology Term Correlations with Generative Adversarial Networks

PLOS ONE

Dear Dr. Rabiee,

Thank you for submitting your manuscript to PLOS ONE. After careful consideration, we feel that it has merit but does not fully meet PLOS ONE’s publication criteria as it currently stands. Therefore, we invite you to submit a revised version of the manuscript that addresses the points raised during the review process.

Two expert reviewers have seen your manuscript and I trust that you will find their comments (see at the bottom of this email) invaluable for preparing a revised version of your work. They highlight several important points - both in terms of presentation as well as technical issues - that should be carefully addressed in a revised manuscript.

We look forward to receiving your revised manuscript.

Kind regards,

Vasilis J Promponas

Academic Editor

PLOS ONE

Journal Requirements:

Reviewers' comments:

Reviewer's Responses to Questions

**Comments to the Author**

1. Is the manuscript technically sound, and do the data support the conclusions?

Reviewer #1: Yes

Reviewer #2: Partly

2. Has the statistical analysis been performed appropriately and rigorously? 

Reviewer #1: Yes

Reviewer #2: N/A

3. Have the authors made all data underlying the findings in their manuscript fully available?

Reviewer #1: No

Reviewer #2: No

4. Is the manuscript presented in an intelligible fashion and written in standard English?

Reviewer #1: Yes

Reviewer #2: Yes

5. Review Comments to the Author

Reviewer #1: The manuscript entitled “PFP-WGAN: Protein Function Prediction by Discovering Gene Ontology Term Correlations with Generative Adversarial Networks” by S. Seyyedsalehi, M. Soleymani, H. Rabiee and M. Mofrad describes PFP-WGAN, a sequence-based tool for protein function prediction. The Authors constructed PFP-WGAN, a conditional generative adversarial network which helps to effectively discover and incorporate term correlations in the gene function annotation process. PFP-WGAN is evaluated and compared to a similar methods reported previously. According to the authors, one of the key advantages of the method is the increased accuracy in predicting more specific function terms associated with the gene.

The manuscript deals with the important problem and tackle it with cutting edge methodology. However, more detailed explanations are necessary to bring the methodology closer to the PlosOne readership.

Major concerns:

Generative Adversarial Network should be mentioned in the Introduction, together with a few lines of description advantages of the approach. The section "Generator Structure and Loss Function" needs to be described in more details.

The authors stated that convolution filters allow for filtering meaningful patterns. Please, explain in more details the filtering procedure and what is meaningful pattern.

Figure 1 needs to be describe in many more details. If possible stages of biological data flow should be presented also. This will facilitate method comprehension to the life scientist.

The Authors should provide information on processing time. How fast PFP-WGAN is in prediction of function for 1000 proteins? How PFP-WGAN compares to similar method such as DEEP-GO?

The Authors should provide an example of both input and output files.

Contrary to what is claimed in Fig. 4 description, FFPred is superior in predicting CC terms. This should be corrected and discussed.

Case studies on how PFP-WGAN outperforms other methods in predicting deep terms in various subontologies would be valuable addition to the manuscript.

The complete training and test sets need to be submitted.

Minor concerns:

Please, use term “child terms” instead of “children”.

Reviewer #2: General Summary: The manuscript by Seyyedsalehi et al. proposes a novel way to train neural protein function predictors. Instead of a standard classification loss, they authors attempt to capture GO term correlations by training an adversarial network. They compare to a multi-label CNN and a multi-task multi-layer perceptron and show that their approach achieves higher Fmax.

The GO is a complicated structure with several constraints such as the true-path rule, term co-occurences and mutual exclusivities, making it difficult to come up with a good “hand-crafted” loss function that also reflects the “realism” of a predicted GO annotation. Therefore, the concept of this study, i.e. learning if a prediction is realistic or not from data is innovative and very interesting. However, I have some concerns about the experiments, mainly the use of appropriate baselines, and the lack of interpretation of the results.

Major comments

1) One thing that troubled me while reading the manuscript is whether this model should be called a GAN. GANs are Generative models whose input is a noise vector (and an extra feature vector in the case of the conditional GANs) and their goal is typically to generate realistic-looking data, such as images, from scratch. Here, there is no noise input (Fig. 1), only a feature vector, and we are dealing with a classic classification task, where an output y is deterministically assigned to an input x. It is trained in an adversarial way, which is the novelty here, but I find that calling it a GAN is a little misleading.

2) In lines 40-51, three previously published methods of exploiting label correlations for function prediction are mentioned (refs 31,32, 28), but the authors do not compare to any of them, because they are “shallow”. I find that this is not a convincing argument not to compare to at least one of them. Comparing to a linear model would give a good baseline for label-correlation-based methods and provide further insight on the superiority of the proposed model. Another relevant linear label dimension reduction model that the authors could compare to is the following:

Bi W., Kwok J. (2011) Multi-label classification on tree-and DAG-structured hierarchies. In: International Conference on Machine Learning

3) Related to that, in lines 88-90 the authors claim that “this is the first time that a deep model is used to explore complex relations and semantic similarities between the GO terms”. This statement is incorrect. The authors should consider the following works: a) GO2vec, Zhong et al., BMC Genomics, 2020, b) Onto2vec, Smaili et al., Bioinformatics, 2018, and c) GOAT, Duong et al., biorxiv, 2020. The last one was specifically modelling GO terms with a deep net for protein function prediction. These are very relevant works and the authors need to benchmark their method against (some of) them.

4) Line 205, The authors mention using a validation set to tune hyperparameters, but in lines 178 and 198 they report “manually” setting the hyperparameters. It has to be clarified what other values were considered for these parameters and how the “manual” decision was made. If the test set is used to decide on these parameters, then the results cannot be trusted.

5) I completely missed the interpretation of the results. Yes, the proposed model works clearly better, but there is no evidence provided that the improvement is indeed due to exploiting label correlations as the authors claim. The first thing that I would like to see is whether the label vectors that are the output of the generator are consistent with the “true path rule”.

6) Again on the interpretation of label correlations: could the authors provide some examples of relationships between labels that their model manages to capture that are not captured by a traditional neural network? For example, there is already evidence that linear GO term correlation models can capture co-occurrence and mutual exclusivity relations between pairs of terms (ref 28). Can the proposed method go beyond this and find more complex relationships?

7) The generator is trained using a weighted sum of a standard cross-entropy loss and the novel adversarial loss proposed in this work. What is the effect of changing the weight parameter lambda_1? What if one makes it really small to only use the cross-entropy? Is then the performance gain lost? And if it is made much larger? Can the model learn to predict functions without the cross-entropy component?

Minor comments

8) The figures are barely readable in the pdf version. The authors should provide higher resolution versions.

9) Broken link that should contain the data (error 404:not found) https://github.com/ictic-bioinformatics/

10) GANs have been previously used in protein function prediction to generate negative examples (Wan and Jones, 2019, biorxiv)

11) Shouldn’t p_m and p_r be flipped in equation 1? Typically in GANs the discriminator has high output for real examples.

12) The DeepGO method is not really modelling label correlations, it is simply enforcing the “true path rule” of the GO graph.

6. PLOS authors have the option to publish the peer review history of their article (what does this mean?). If published, this will include your full peer review and any attached files.

Reviewer #1: No

Reviewer #2: No

---

## [Author Response · Author response to Decision Letter 0]

30 Aug 2020

Dear Editor,

The authors would like to thank you for providing the opportunity to respond to the comments. The manuscript has been completely revised based on the reviewers’ comments. We also adjusted images carefully according to rules of P¬¬los One. In the revised manuscript, we have addressed all editor’s and reviewers’ concerns. Detailed response and discussion are provided in the following pages.

All the changes to the original manuscript, including new references, are highlighted in yellow in the marked-up copy of the manuscript. The authors’ responses appear in blue color below.

Sincerely Yours,

Hamid R. Rabiee

The corresponding author

August 30, 2020 

Reviewer 1

Comments to the Author

The manuscript entitled “PFP-WGAN: Protein Function Prediction by Discovering Gene Ontology Term Correlations with Generative Adversarial Networks” by S. Seyyedsalehi, M. Soleymani, H. Rabiee and M. Mofrad describes PFP-WGAN, a sequence-based tool for protein function prediction. The Authors constructed PFP-WGAN, a conditional generative adversarial network which helps to effectively discover and incorporate term correlations in the gene function annotation process. PFP-WGAN is evaluated and compared to a similar methods reported previously. According to the authors, one of the key advantages of the method is the increased accuracy in predicting more specific function terms associated with the gene. 

The manuscript deals with the important problem and tackle it with cutting edge methodology. However, more detailed explanations are necessary to bring the methodology closer to the PlosOne readership.

[Authors’ response:] The authors would like to thank the reviewer for his/her in-depth analysis and useful comments. Below we have listed the issues raised by you and addressed them as best as we could, and we have revised the manuscript in accordance with the reviewer’s comments, as needed.

Major concerns:

1. Generative Adversarial Network should be mentioned in the Introduction, together with a few lines of description advantages of the approach. The section "Generator Structure and Loss Function" needs to be described in more details.

The authors stated that convolution filters allow for filtering meaningful patterns. Please, explain in more details the filtering procedure and what is meaningful pattern.

[Authors’ response:] We added a description of GANs and their successes and applications to the introduction between Lines 99-113. 

According to the reviewer’s comment, we explained "Generator Structure and Loss Function" in more detail and explained filtering procedure and meaningful patterns in Lines 185-191 as follows:

“For the next layer, we use 32 one-dimensional convolution filters which extract meaningful patterns from the sequence of amino acids. After the training process, each filter is responsible for detecting a specific pattern. By patterns we mean the existence of special sequence of amino acids in specific positions. Meaningful patterns are those which are correlated with different GO terms. When a sequence is passed through these filters an activation map, which shows the matching score between patterns and input sequence, is obtained.”

2. Figure 1 needs to be described in many more details. If possible stages of biological data flow should be presented also. This will facilitate method comprehension to the life scientist.

[Authors’ response:] Thank you for this valuable suggestion. We added more details about the proposed method to Fig. 1. We also described it in more details in its caption and section "Generator Structure and Loss Function" in Lines 185 – 200. 

3. The Authors should provide information on processing time. How fast PFP-WGAN is in prediction of function for 1000 proteins? How PFP-WGAN compares to similar method such as DEEP-GO?

[Authors’ response:] Thank you for indicating this point. We added a comparison between the prediction time of PFP-WGAN and DEEPGO in Table 2 of the revised manuscript. There is not a considerable difference between prediction times. However, as the number of terms is increased, the growth of averaged prediction time in DeepGO is more than PFP-WGAN. Therefore, PFP-WGAN is more scalable for predicting a large number of GO terms simultaneously. 

4. The Authors should provide an example of both input and output files.

[Authors’ response:] A sample of input and output files are provided in: http://git.dml.ir/seyyedsalehi/PFP-WGAN

5. Contrary to what is claimed in Fig. 4 description, FFPred is superior in predicting CC terms. This should be corrected and discussed.

[Authors’ response:] Thank you for mentioning this mistake in reporting the results. Although the FFPred and our method have comparable performance on CC Terms, after considering comments of the reviewers, we corrected the report of the results and now the previous claim is correct as can be seen in Fig. 5.

6. Case studies on how PFP-WGAN outperforms other methods in predicting deep terms in various subontologies would be valuable addition to the manuscript.

[Authors’ response:] Thank you for the valuable suggestion. In Fig. 4, we show the F1 measure obtained for GO terms by PFP-WGAN and DeepGO as a function of available positive training samples. As mentioned in the manuscript in Lines 344-349, the improvement obtained for rare terms is more considerable. As deeper terms have less number of positive samples, it confirms the ability of PFP-WGAN to improve F1 measure for them.

7. The complete training and test sets need to be submitted.

[Authors’ response:] All data and code are now available at: 

http://git.dml.ir/seyyedsalehi/PFP-WGAN.

I addition, as we have mentioned in the manuscript, the data for the first experiment is the original data of DeepGO [11], which is available at:

http://deepgo.bio2vec.net/data/deepgo/data.tar.gz, and the data which is used in the second experiment is the same as the one used in MTDNN [5], available from:

http://bioinf.cs.ucl.ac.uk/downloads/mtdnn.

Minor concerns:

8. Please, use term “child terms” instead of “children”.

[Authors’ response:] Thanks for your suggestion. We replaced all “children” with “child terms” as requested.

Reviewer 2

Comments to the Author

General Summary: The manuscript by Seyyedsalehi et al. proposes a novel way to train neural protein function predictors. Instead of a standard classification loss, the authors attempt to capture GO term correlations by training an adversarial network. They compare to a multi-label CNN and a multi-task multi-layer perceptron and show that their approach achieves higher Fmax.

The GO is a complicated structure with several constraints such as the true-path rule, term co-occurrences and mutual exclusivities, making it difficult to come up with a good “hand-crafted” loss function that also reflects the “realism” of a predicted GO annotation. Therefore, the concept of this study, i.e. learning if a prediction is realistic or not from data is innovative and very interesting. However, I have some concerns about the experiments, mainly the use of appropriate baselines, and the lack of interpretation of the results.

[Authors’ response:] The authors would like to thank the reviewer for his/her in-depth analysis and useful comments. The comments are answered accordingly as follows.

Major concerns:

1. One thing that troubled me while reading the manuscript is whether this model should be called a GAN. GANs are Generative models whose input is a noise vector (and an extra feature vector in the case of the conditional GANs) and their goal is typically to generate realistic-looking data, such as images, from scratch. Here, there is no noise input (Fig. 1), only a feature vector, and we are dealing with a classic classification task, where an output y is deterministically assigned to an input x. It is trained in an adversarial way, which is the novelty here, but I find that calling it a GAN is a little misleading.

[Authors’ response:] We agree with the respected reviewer that the generator of the main GAN generally needs to have a noise signal as an input. But it is common in the literature to call the type of conditional models similar to our proposed model as GAN, too. For example, the following works refer to their method as GAN while their generator’s inputs do not include noise:

R1. Isola, Phillip, et al. "Image-to-image translation with conditional adversarial networks." Proceedings of the IEEE conference on computer vision and pattern recognition. 2017. 

R2. Zhu, Jun-Yan, Taesung Park, Phillip Isola, and Alexei A. Efros. "Unpaired image-to-image translation using cycle-consistent adversarial networks." In Proceedings of the IEEE international conference on computer vision, pp. 2223-2232. 2017.

Without the noise signal, the conditional generator learns a deterministic mapping between the input and the output space (i.e. it generates a sample from the output space given the input). As mentioned in (R1), past conditional GANs have acknowledged using Gaussian noise but in some situations the generator learns to simply ignore it. Therefore, similar to the work in (R1) we provide the noise in the form of a dropout layer (as it is shown in Fig 1).

2. In lines 40-51, three previously published methods of exploiting label correlations for function prediction are mentioned (refs 31, 32, 28), but the authors do not compare to any of them, because they are “shallow”. I find that this is not a convincing argument not to compare to at least one of them. Comparing to a linear model would give a good baseline for label-correlation-based methods and provide further insight on the superiority of the proposed model. Another relevant linear label dimension reduction model that the authors could compare to is the following:

Bi W., Kwok J. (2011) Multi-label classification on tree-and DAG-structured hierarchies. In: International Conference on Machine Learning.

[Authors’ response:] Thanks for this valuable comment. We introduced “Multi-label classification on tree-and DAG-structured hierarchies” in the introduction in Lines 44-47 and reported its result in the second experiment. Because this paper does not work on sequences directly and needs a feature vector as input, we did the second experiment on it. The result is shown in Fig.5 of the revised manuscript.

3. Related to that, in lines 88-90 the authors claim that “this is the first time that a deep model is used to explore complex relations and semantic similarities between the GO terms”. This statement is incorrect. The authors should consider the following works: a) GO2vec, Zhong et al., BMC Genomics, 2020, b) Onto2vec, Smaili et al., Bioinformatics, 2018, and c) GOAT, Duong et al., biorxiv, 2020. The last one was specifically modelling GO terms with a deep net for protein function prediction. These are very relevant works and the authors need to benchmark their method against (some of) them.

[Authors’ response:] This comment is highly appreciated. We omitted the sentence “this is the first time that a deep model is used to explore complex relations and semantic similarities between the GO terms” and added those recent works to the paper (references [34], [36] and [37]). We review GO2vec in Lines 53-58 and Onto2vec in Lines 58-63. However, it is worth mentioning that the methods like Onto2vec [34] are not deep, although they are mentioned as deep methods in some studies. Moreover, the goal of GO2vec and Onto2vec is different from ours. They use the representations of GO terms to compute semantic similarity between GO terms and consequently the functional similarity between proteins. However, since they find a feature vector for proteins based on their annotations, their method cannot be applied to new proteins without GO annotations. Therefore, their method cannot directly be used for protein function prediction. 

GOAT proposes a model to annotate proteins by GO and as you have mentioned in your comment, it is completely relevant to our work. We review it in the introduction of revised manuscript in Lines 93-98. However, as this work is proposed recently we found some ambiguities and inconsistencies in its code and paper, and we could not compare our method with it during the revision time. In Lines 217-220 of their bioRxiv manuscript, they explain that they use the dataset of DeepGO and omit proteins without annotations. We omitted such proteins from the original DeepGO dataset nonetheless we could not obtain the dataset provided by them in: 

https://drive.google.com/drive/folders/1cuO2WtfZX2_vyk0Z8S7suYlYzpbwPl-j

In addition, the size of the dataset they explain in their paper in Lines 217-220 is completely different from the size of the dataset provided by them in the above link, original DeepGO dataset, and the original DeepGO dataset filtered by un-annotated proteins (as mentioned in their paper). Moreover, their code is not easy-to-use and we could not obtain its results on our datasets. 

4. Line 205, The authors mention using a validation set to tune hyperparameters, but in lines 178 and 198 they report “manually” setting the hyperparameters. It has to be clarified what other values were considered for these parameters and how the “manual” decision was made. If the test set is used to decide on these parameters, then the results cannot be trusted.

[Authors’ response:] Thank you for pointing out this inconsistency. As mentioned in section “Training and Optimization”, we use 20% of the training data for validation to tune hyper-parameters. We added details in Lines 212-214 and 233-235, and emphasized using validation set to tune hyper-parameters. 

5. I completely missed the interpretation of the results. Yes, the proposed model works clearly better, but there is no evidence provided that the improvement is indeed due to exploiting label correlations as the authors claim. The first thing that I would like to see is whether the label vectors that are the output of the generator are consistent with the “true path rule”.

[Authors’ response:] The main idea of our work is adding a discriminator to a deep annotation model to judge it by observing valid GO annotations in SwissProt. As explained in the manuscript, this discriminator learns a distribution over GO annotation space. If it is trained successfully, it can reflect the correlation between GO terms by giving high scores to annotations with respect to co-occurrence and mutual exclusivity relations between GO terms. The discriminator captures these relations automatically by observing valid GO annotations. Therefore, the superiority of the model which is obtained by this block is obviously the result of learning GO correlations. 

To evaluate the consistency of our outputs with the true-path-rule, we define (Lines 288-291) and report the TPR score in Lines 349-351 and 377-379. TPR shows the expected number of conflicts in the annotation of a protein. A conflict is a pair of terms that a protein is annotated by just one of them which is the grandchild of another one. The TPR score of the PFP-WGAN on the first dataset is 0.78, 0.01 and 0.11 for BP, MF and CC branches, respectively. In addition, the total number of grandchild and ancestor pairs of the tree of each branch is 8323, 3266 and 3106. In the second dataset, this score is 0.56, 0.11 and 0.12 for BP, MF and CC branches, respectively. In addition, the total number of grandchild and ancestor pairs of the tree of each branch is 292, 84 and 44.

6. Again on the interpretation of label correlations: could the authors provide some examples of relationships between labels that their model manages to capture that are not captured by a traditional neural network? For example, there is already evidence that linear GO term correlation models can capture co-occurrence and mutual exclusivity relations between pairs of terms (ref. 28). Can the proposed method go beyond this and find more complex relationships?

[Authors’ response:] In the proposed model, we exploit a conditional GAN for protein annotation and learning GO term relations simultaneously. It means the discriminator learns a distribution over GO annotations conditioned on the protein sequence. Therefore, it is also able to extract correlations which are valid for specific sequence patterns. 

For example, in general it could be possible that there is no co-occurrence relation between two GO terms, but for protein sequences with special amino acids in special positions these two terms occurred concurrently. 

Moreover, the proposed method is not limited to checking binary relations at all. It can discover complex relations between multiple GO terms, thanks to its discriminator which processes whole functions together using a deep network. 

To evaluate the ability of method in capturing GO correlations which exist in the training dataset, we added a new analysis to the revised version of the manuscript in Lines 364-373. For each of training and testing dataset, we calculate a binary heatmap which shows whether two GO terms appear in at least one protein, simultaneously or not. Therefore, these GO terms are as consistent, such that a protein can be annotated by both of them simultaneously. We obtain this heatmap for the training dataset where predictions obtained by PFP-WGAN and prediction obtained by omitting the adversarial loss from PFP-WGAN. Results of this analysis is provided in the revised manuscript. As shown, The MSE between the heatmap of PFP-WGAN and training dataset is less than the MSE between the simple deep network and training dataset. This fact confirms that PFP-WGAN has more ability to explore and utilize these relations from the training data. 

7. The generator is trained using a weighted sum of a standard cross-entropy loss and the novel adversarial loss proposed in this work. What is the effect of changing the weight parameter lambda_1? What if one makes it really small to only use the cross-entropy? Is then the performance gain lost? And if it is made much larger? Can the model learn to predict functions without the cross-entropy component?

[Authors’ response:] Both of these losses are necessary for our training process and have different information. For each input sample the binary cross-entropy compares the obtained output by the ground truth for this specific sample. But the adversarial loss judges this pair (the input sequence and obtained output) considering all inputs and their ground truths. In fact, the discriminator checks if the obtained result has features of a valid annotation or not. We added the sensitivity analysis of PFP-WGAN on this parameter in Lines 374-376 and Fig. 6. As shown in this figure, by omitting each of these losses the performance is being degraded. 

Minor concerns:

8. The figures are barely readable in the pdf version. The authors should provide higher resolution versions.

[Authors’ response:] Thank you for indicating this issue. We have provided higher resolution versions of figures in the revised version.

9. Broken link that should contain the data (error 404:not found) https://github.com/ictic-bioinformatics/

[Authors’ response:] All data and code are now available at: 

http://git.dml.ir/seyyedsalehi/PFP-WGAN.

In addition, as we have mentioned in the manuscript, the data for the first experiment is the original data of DeepGO [11] which is available from: 

http://deepgo.bio2vec.net/data/deepgo/data.tar.gz, And the data which is used in second experiment is the same as one used in MTDNN [5] available from: 

http://bioinf.cs.ucl.ac.uk/downloads/mtdnn.

10. GANs have been previously used in protein function prediction to generate negative examples (Wan and Jones, 2019, biorxiv).

[Authors’ response:] The comment is highly appreciated. The mentioned reference has been carefully reviewed in Lines 111-113 and added to the references (reference [45]). However, this work is completely different from ours. It utilizes the GAN to generate samples (i.e. protein feature vectors) and perform data augmentation. It then uses them as training samples to increase the accuracy of a classifier which predict protein’s functions. But we exploit an adversarial approach (called GAN) to extract GO term correlations. We do not generate any protein and instead we try to learn the mapping between the proteins and the functions in an adversarial manner (by considering the adversarial loss). 

11. Shouldn’t p_m and p_r be flipped in equation 1? Typically in GANs the discriminator has high output for real examples.

[Authors’ response:] Thanks for mentioning this mistake in notations. Symbols p_r and p_m are now flipped in Eq. 1, and all other equations which are affected. 

12. The DeepGO method is not really modelling label correlations, it is simply enforcing the “true path rule” of the GO graph.

[Authors’ response:] Thank you for this comment. We highlighted this fact in the introduction part where we have reviewed DeepGO (Lines 80-82). 

DeepGO just adds a maximization layer as the final step of its network. Authors in DeepGO attempt to impose this structural information during their training phase by calculating the loss function for the output of the maximization layer. However, as we discussed in the introduction part, the model cannot provide a sufficient gradient.

---

## [Decision Letter · Decision Letter 1]

22 Oct 2020

PONE-D-20-05418R1

PFP-WGAN: Protein Function Prediction by Discovering Gene Ontology Term Correlations with Generative Adversarial Networks

PLOS ONE

Dear Dr. Rabiee,

Thank you for submitting your manuscript to PLOS ONE. After careful consideration, we feel that it has merit but does not fully meet PLOS ONE’s publication criteria as it currently stands. Therefore, we invite you to submit a revised version of the manuscript that addresses the points raised during the review process.

We look forward to receiving your revised manuscript.

Kind regards,

Alexandros Iosiﬁdis

Academic Editor

PLOS ONE

Additional Editor Comments (if provided):

Both reviewers agree that the paper has merits. Please address the comments provided by Reviewer 2 and provide a point-to-point response letter in your revision.

Reviewers' comments:

Reviewer's Responses to Questions

**Comments to the Author**

1. If the authors have adequately addressed your comments raised in a previous round of review and you feel that this manuscript is now acceptable for publication, you may indicate that here to bypass the “Comments to the Author” section, enter your conflict of interest statement in the “Confidential to Editor” section, and submit your "Accept" recommendation.

Reviewer #1: All comments have been addressed

Reviewer #2: (No Response)

2. Is the manuscript technically sound, and do the data support the conclusions?

Reviewer #1: Yes

Reviewer #2: Partly

3. Has the statistical analysis been performed appropriately and rigorously? 

Reviewer #1: Yes

Reviewer #2: N/A

4. Have the authors made all data underlying the findings in their manuscript fully available?

Reviewer #1: No

Reviewer #2: Yes

5. Is the manuscript presented in an intelligible fashion and written in standard English?

Reviewer #1: Yes

Reviewer #2: Yes

6. Review Comments to the Author

Reviewer #1: The authors have satisfyingly addressed my comments. However, data are still not accessible as link http://git.dml.ir/seyyedsalehi/PFP-WGAN is not active.

This should be fixed before manuscript is accepted for publishing.

Reviewer #2: The authors have addressed most of my comments, but some minor points remain:

1. Table 3 shows that the WGAN is better at capturing (linear) co-occurrence relations between terms. In lines 124-125 the authors state that this model ‘ is able to model more complicated and higher level correlations that are not necessarily available in the current DAG model’. The results do not show any evidence of ability to model higher order relations, so this statement should be removed or changed to something like ‘is able to mode co-occurence relations that are not necessarily available in the current DAG model’

2. In the text authors mention the use of dropout to avoid overfitting, but from their answers to comment 1 it seems they also use dropout to provide stochasticity for the generator. If this is the case, it should be mentioned in the manuscript

3. The authors should explain the TPR score better: preferably provide a formula and explain what it means

4. Typo in line 56

5. The figure definition is better, but still not publication-quality in my opinion. The editorial stuff can perhaps provide information on how to generate high-quality figures

7. PLOS authors have the option to publish the peer review history of their article (what does this mean?). If published, this will include your full peer review and any attached files.

Reviewer #1: No

Reviewer #2: No

---

## [Author Response · Author response to Decision Letter 1]

30 Oct 2020

Reviewer 1

Comments to the Author:

The authors have satisfyingly addressed my comments. However, data are still not accessible as link http://git.dml.ir/seyyedsalehi/PFP-WGAN is not active.

This should be fixed before manuscript is accepted for publishing.

[Authors’ response:] Thank you. Possibly the link was unreachable at the moment you have checked it, because of some temporary technical issues. The problem is solved now and the link is working properly. 

In addition to link, the original data is alternatively available from the previous works that are referenced in the paper, i.e. Ref[5] and Ref[11].

Reviewer 2

Comments to the Author:

The authors have addressed most of my comments, but some minor points remain:

1. Table 3 shows that the WGAN is better at capturing (linear) co-occurrence relations between terms. In lines 124-125 the authors state that this model ‘ is able to model more complicated and higher level correlations that are not necessarily available in the current DAG model’. The results do not show any evidence of ability to model higher order relations, so this statement should be removed or changed to something like ‘is able to model co-occurrence relations that are not necessarily available in the current DAG model’

[Authors’ response:] Thank you for indicating this point. We changed this sentence based on your suggestion in lines 123-125. 

2. In the text authors mention the use of dropout to avoid overfitting, but from their answers to comment 1 it seems they also use dropout to provide stochasticity for the generator. If this is the case, it should be mentioned in the manuscript

[Authors’ response:] Thank you for this valuable comment. In line 183 we mentioned that we add the dropout layer to have a stochastic generator in addition to avoid overfitting.

3. The authors should explain the TPR score better: preferably provide a formula and explain what it means

[Authors’ response:] Thank you for your suggestion. We provide a formula for TPR in equation (10) and we described it in lines 288-295.

4. Typo in line 56

[Authors’ response:] “grpah” is changed to graph.

5. The figure definition is better, but still not publication-quality in my opinion. The editorial stuff can perhaps provide information on how to generate high-quality figures.

[Authors’ response:] We tried to follow the rules provided on the following page:

https://journals.plos.org/plosone/s/figures

All of our original figures that we submit to PLOS ONE are in TIF format with 300 dpi resolution. All the dimensions are also consistent with the mentioned rules. However, the draft file you may have received has lower quality images. One reason could be that this draft file is auto-generated. We follow up with the editorial staff instructions to make sure the quality of figures are as expected in the final version of the paper.

---

## [Decision Letter · Decision Letter 2]

10 Dec 2020

PFP-WGAN: Protein Function Prediction by Discovering Gene Ontology Term Correlations with Generative Adversarial Networks

PONE-D-20-05418R2

Dear Dr. Rabiee,

We’re pleased to inform you that your manuscript has been judged scientifically suitable for publication and will be formally accepted for publication once it meets all outstanding technical requirements.

Kind regards,

Alexandros Iosiﬁdis

Academic Editor

PLOS ONE

Additional Editor Comments (optional):

The Reviewers are satisfied with the current version of the paper. Congratulations on the acceptance of your paper.

Reviewers' comments:

Reviewer's Responses to Questions

**Comments to the Author**

1. If the authors have adequately addressed your comments raised in a previous round of review and you feel that this manuscript is now acceptable for publication, you may indicate that here to bypass the “Comments to the Author” section, enter your conflict of interest statement in the “Confidential to Editor” section, and submit your "Accept" recommendation.

Reviewer #1: (No Response)

Reviewer #2: All comments have been addressed

2. Is the manuscript technically sound, and do the data support the conclusions?

Reviewer #1: Yes

Reviewer #2: Yes

3. Has the statistical analysis been performed appropriately and rigorously? 

Reviewer #1: Yes

Reviewer #2: Yes

4. Have the authors made all data underlying the findings in their manuscript fully available?

Reviewer #1: Yes

Reviewer #2: Yes

5. Is the manuscript presented in an intelligible fashion and written in standard English?

Reviewer #1: Yes

Reviewer #2: Yes

6. Review Comments to the Author

Reviewer #1: (No Response)

Reviewer #2: (No Response)

7. PLOS authors have the option to publish the peer review history of their article (what does this mean?). If published, this will include your full peer review and any attached files.

Reviewer #1: No

Reviewer #2: No

---

## [Editor Report · Acceptance letter]

18 Dec 2020

PONE-D-20-05418R2 

PFP-WGAN: Protein Function Prediction by Discovering Gene Ontology Term Correlations with Generative Adversarial Networks  

Dear Dr. Rabiee:

I'm pleased to inform you that your manuscript has been deemed suitable for publication in PLOS ONE. Congratulations! Your manuscript is now with our production department. 

Kind regards, 

on behalf of

Dr. Alexandros Iosiﬁdis 

Academic Editor

PLOS ONE